# Microbial community organization designates distinct pulmonary exacerbation types and predicts treatment outcome in cystic fibrosis

Stefanie Widder [1] ✉, Lisa A. Carmody[2], Kristopher Opron[3], Linda M. Kalikin[2], Lindsay J. Caverly[2] & John J. LiPuma[2]

Polymicrobial infection of the airways is a hallmark of obstructive lung diseases such as cystic fibrosis (CF), non-CF bronchiectasis, and chronic obstructive pulmonary disease. Pulmonary exacerbations (PEx) in these conditions are associated with accelerated lung function decline and higher mortality rates. Understanding PEx ecology is challenged by high inter-patient variability in airway microbial community profiles. We analyze bacterial communities in 880 CF sputum samples collected during an observational prospective cohort study and develop microbiome descriptors to model community reorganization prior to and during 18 PEx. We identify two microbial dysbiosis regimes with opposing ecology and dynamics. Pathogen-governed PEx show hierarchical community reorganization and reduced diversity, whereas anaerobic bloom PEx display stochasticity and increased diversity. A simulation of antimicrobial treatment predicts better efficacy for hierarchically organized communities. This link between PEx, microbiome organization, and treatment success advances the development of personalized clinical management in CF and, potentially, other obstructive lung diseases.

Obstructive lung diseases, such as cystic fibrosis (CF), non-CF bronchiectasis, and chronic obstructive pulmonary disease (COPD), are characterized by chronic polymicrobial bacterial infection of the airways. Intermittent increases in signs and symptoms of respiratory dysfunction, so-called pulmonary exacerbations (PEx), are associated with lung disease progression and mortality in these conditions[1–3]. Despite their importance, the pathophysiologic events underlying PEx are unclear but generally believed to involve transient perturbation of host-microbial dynamics in the airways. Management of these events typically involves frequent, often aggressive, antibiotic treatment, which is intended to decrease bacterial burden and blunt host

inflammatory response that contributes to lung pathology. This care carries considerable cost and treatment burden and is limited by drug toxicity and ever-increasing antimicrobial drug resistance[4]. In CF, therapies that modulate the activity of the dysfunctional CF transmembrane conductance regulator (CFTR), the primary cellular defect in CF, have reduced the frequency of PEx for many, but not all, people with CF[5,6]. Thus, a better understanding of PEx remains a high priority in efforts to improve care and enhance the quality of life for persons with obstructive lung conditions[7].

Dysbiosis defines disease-associated alterations of the microbiome that affect the taxonomic composition as well as the functional

[1]Department of Medicine 1, Research Division Infection Biology, Medical University of Vienna, 1090 Vienna, Austria. [2]Department of Pediatrics, University of Michigan Medical School, Ann Arbor, MI 48109, USA. [3]Department of Internal Medicine, University of Michigan Medical School, Ann Arbor, MI 48109, USA. ✉ e-mail: stefanie.widder@meduniwien.ac.at

activity of the microbial community[8]. This serves as an umbrella term for a variety of non-exclusive community characteristics, including diversity loss, symbiont loss, or pathobiont blooms. As such, the label dysbiosis may be of limited applicability in describing microbial dynamics in chronic obstructive lung diseases insofar as the pulmonary microbiome in these conditions displays a markedly different ecology from that in healthy lungs, and can be considered, by definition, to represent a dysbiotic state[9,10]. Nevertheless, given that a pathologic microbiome persists even during periods of relative clinical stability, a better characterization of its reorganization patterns to classify (relative) pulmonary dysbiosis into distinct types could provide opportunities for improved management of PEx in chronic pulmonary conditions[8,10,11].

The search, in cross-sectional studies, for common motifs in microbial community processes that drive PEx, particularly in CF, has been hampered by subject-specific microbiome configurations. Longitudinal sampling strategies have revealed highly individual taxonomic profiles with context-dependent metabolic activities and signaling in numerous studies[12–15]. Unpredictable and ill-defined onset of PEx, as well as personalized antimicrobial treatment schemes to manage PEx, further complicate analyses[16]. A strategy that is capable of consolidating process communalities against the background of natural case variability is therefore required[17].

A number of studies on microbial community networks have found that the precise configuration of dependencies among members define their community role, as well as the dynamical behavior of the microbiome[18–20]. Moreover, the formation of network clusters (i.e., the coexistence of microbial sub-communities) modulates robustness to external perturbation including antimicrobial therapy[21,22]. Recently, time evolution of gut, vaginal, and oral microbiomes were modeled using alternative community descriptors anchored in information theory[23]. Switching patterns of microbiota that emerged as tradeoff between perturbation, accessible niches and internal forces were identified. The impact of complex community organization on medically relevant microbial behaviors such as pathogen virulence or resilience to antimicrobial therapy is understudied and remains largely unexplored for clinical applications.

In this study, we hypothesized that discernable microbial patterns exist that delimit different types of PEx in CF. We developed non-standard descriptors that aggregated ecological and compositional properties of the CF lung microbiome and used these to identify PEx types with communal patterns. We then analyzed the organization of the CF microbiome in these backgrounds and revealed two fundamental dysbiosis states: a hierarchical community reorganization controlled by the dominant pathogen, and a stochastic reorganization with blooming anaerobic taxa and high taxonomic turnover. Of note, the behavior of a focal pathogen was markedly different with different community hierarchy. Lastly, we modeled targeted antimicrobial treatment on data-inferred co-occurrence networks and observed that distinct community organizations significantly determined treatment outcomes.

## Results and discussion
### Compositional characterization of PEx time series
We aggregated a collection of 880 sputum samples from 11 adults with CF, comprising 18 PEx time series. The characteristics of the study subjects and sputum samples are provided in Table 1, and as supplementary information in the Source Data file/Sample Data; sample inclusion criteria are provided in Tables S1 and S2. Subjects and sputum samples were chosen from a larger dataset that had been generated during the course of a long-term observational study[15,24–27]. Subjects were selected from this larger dataset based on the availability of near-daily sputum samples (i.e., samples available from at least 60% of days) that spanned periods of clinical stability culminating with a PEx that prompted antibiotic treatment by the subject's care

team. More specifically, samples in each PEx time series were collected from 60 days prior to one day prior to the initiation of antibiotic treatment for PEx (Fig. S1). The time frame of 60 days prior to the start of PEx treatment was selected to accommodate potential changes in the lung microbiome preceding symptom onset, together with changes occurring during the acute PEx phase. As we have done in previous studies[15,24,28–33], samples were further characterized based on the subject's clinical state at the time of collection: baseline samples were those collected between 60 and 15 days prior to the start of antibiotic treatment; exacerbation samples were collected between 14 and one day prior to the start of antibiotic treatment. Neither samples obtained during acute antibiotic treatment for PEx (treatment samples) nor within three weeks after PEx treatment stopped (recovery samples) were included in this analysis. Chronic (maintenance) antibiotic therapies such as inhaled tobramycin and aztreonam, and oral azithromycin used on each day were recorded. A mean of 49 (SD, 7.3) sputum samples were analyzed per PEx time series.

Identifying common microbiological features of PEx in CF is challenged by the pronounced subject specificity of the lung microbiome, which typically overshadows potential communalities. Accordingly, we identified 1949 amplicon sequence variants (ASVs) among the 880 samples, with only eight ASVs present in every subject. For further analyses, the dataset was denoised to 194 core ASVs by removing taxa present with an average relative abundance below 0.0075%. To quantify the degree of subject specificity in the dataset, a PERMANOVA test was performed to calculate the effect sizes of clinical and demographic covariates on data variance. Covariates included subject, subject age, subject sex, clinical state (baseline health or exacerbation of symptoms[34]), and zygosity of the *cftr* F508del allele. As expected, we found that individual subject was a strong predictor for ASV covariance (PERMANOVA, $\omega^2 = 0.51$, p < 0.001), followed, to lesser effects, by age (PERMANOVA, $\omega^2 = 0.023$, p < 0.001) and clinical state (PERMANOVA, $\omega^2 = 0.003$, p < 0.003).

### Identifying distinct PEx types using non-standard descriptors
To reduce the subject-specific microbiome bias, we abandoned ASV composition as the *sole* sample descriptor, assembling the 194 core ASVs into five higher-order groups. The first group comprised conventional CF pathogens (*Pseudomonas, Staphylococcus, Burkholderia, Haemophilus, Achromobacter*, and *Stenotrophomonas*) based on the prominent role these species are believed to play in CF lung disease[35,36]. Three groups were categorized reflecting species oxygen requirement for growth[37], considering that the CF lung microbiome is strongly conditioned by local oxygen gradients: strictly aerobic, strictly anaerobic, and facultatively anaerobic. The fifth group comprised uncultivated taxa with unknown oxygen requirements. A detailed list of how ASVs distribute across the groups is provided as supplementary information in the Source Data file.

Building on these five ASV categories, we assembled the following non-standard descriptors for every sputum sample: (i) the ratio of CF pathogens to strict anaerobes, (ii) the relative abundance of the most abundant CF pathogen, (iii) the Shannon diversity index of the core ASVs, (iv) the Chao1 richness of the core ASVs, and (v) a community typing using Dirichlet multinomial mixtures (DMM). The DMM model was implemented using the five ASV groups as input and identified six community classes. Two DMM community classes were dominated by CF pathogens, three by anaerobes, and one by facultatively anaerobic organisms. Community classes, selection of Dirichlet components, class distribution over the cohort and class-wise sample compositions are presented in Fig. S2.

Using this suite of descriptors, we implemented a second PERMANOVA model and found that the variance explained by this model was reduced compared to that based solely on ASVs (PERMANOVA, $R^2 = 0.62$ and $R^2 = 0.78$, respectively). Most importantly, the effect size of subject bias decreased by 51% (Fig. 1A). Subsequently, data were

**Table 1 | Study subjects and sputum samples**

| Subjects | | 11 |
|---|---|---|
| Mean age in years (range) | | 35.3 (22–52) |
| No. per sex | Female | 5 |
| | Male | 6 |
| No. per CFTR mutation | F508del homozygous | 7 |
| | F508del heterozygous | 3 |
| | Other | 1 |
| No. per disease stage | Early (ppFEV$_1$ ≥ 70%) | 6 |
| | Intermediate (70% > ppFEV$_1$ ≥ 40%) | 3 |
| | Advanced (ppFEV$_1$ < 40%) | 2 |
| **PEx time series** | | 18 |
| Mean PEx per subject (range) | | 1.6 (1–3) |
| No. PEx per subject A-K | | 3, 1, 1, 1, 3, 1, 1, 1, 3, 1, 2 |
| **Samples** | | 880 |
| Mean per subject (range) | | 80.0 (37–168) |
| No. per subject A-K | | 126, 53, 38, 50, 143, 37, 51, 51, 168, 56, 107 |
| Mean per PEx time series (range) | | 48.9 (36–59) |
| No. per PEx time series 1–18 | | 40, 36, 50, 53, 38, 50, 55, 48, 40, 37, 51, 51, 56, 53, 59, 56, 54, 53 |
| No. per clinical state | Baseline | 674 |
| | Exacerbation | 206 |
| **Antibiotic maintenance therapy** | | |
| No. of samples per no. of subjects | Azithromycin oral | 133/7 |
| | Aztreonam inhaled | 292/8 |
| | Others | 153/6 |
| | None | 302/9 |

ordinated independent of clinical state using principal component analysis (Fig. 1B), and the first three principal components were used to group similar samples. K-mer clustering identified three distinct PEx clusters or types using $\chi^2$ statistics (Fig. 1C, D). ASVs in PEx clusters are detailed in the Source Data file.

Having identified three robust clusters, hereafter referred to as PEx types, we next analyzed the distribution of DMM communities among these. PEx Type 1 (hereafter called PAT) comprised communities dominated by conventional CF pathogens, including *Pseudomonas*, *Burkholderia*, *Achromobacter*, *Haemophilus, Staphylococcus*, and *Stenotrophomonas*. PEx Type 2 (AN1) and Type 3 (AN2), on the other hand, were driven by three distinct anaerobic community configurations. These results suggested that species-sorting occurred in subjects' lungs according to oxygen requirements[38]. Importantly, PEx proceeded in both aerobic and anaerobic communities.

To assign subjects and their PEx time series to a single PEx type, we performed Spearman's rank association (Figs. S3A and S3B). Two time series were excluded from further analyses due to inconclusive association to a single type (time series 9, 12). The reduced number of 789 samples distributed as 286, 254, and 249; the number of subjects as 4, 3, and 4; and the number of PEx as 6, 5, and 5 to PEx types PAT, AN1, and AN2, respectively (Fig. S3C and S3D). We found that sample association with PEx type PAT was remarkably stable both at the level of individual PEx time series (60 days), as well as with subjects over

time. On the contrary, more transition events were observed between PEx types AN1 and AN2 (Fig. S4A and S4B). Overall, subjects showed a tendency to persist either in PAT or in AN1 or AN2 despite recurrent antibiotic treatment between time series (treatment samples excluded, Fig. S4C).

In summary, aggregated measures of sample diversity, ecology and function were used to reduce the organism-driven subject bias and group PEx trajectories with similar properties. We identified three communal PEx types among subjects, termed PAT, AN1 and AN2, that displayed distinguishable microbiomes.

**Temporal behavior of microbiomes in distinct exacerbation regimes**

We studied the configuration of the lung microbiota in and between the identified PEx types and modeled common reorganization patterns over time as the community proceeded towards the start of PEx treatment. To elucidate underlying ecological processes, we first asked whether PEx types could be simply explained by the DMM community classes, i.e., different community compositions[28,39] and whether oxygen availability could motivate shifts in microbiome configurations[40]. We assessed the distribution and temporal change of DMM community classes previously modeled from coarse-grained ASV groups (Fig. 2A, S2). Unexpectedly, no significant temporal evolution of DMM communities was observed within PEx types, indicating that the overall proportions of pathogens, anaerobes, facultative anaerobes, and aerobes persisted over most of the PEx cycle with few exceptions. These sporadic shifts occurred only between comparable community classes, i.e., due to continuous transitions (increase or decrease) of taxonomic groups.

Several studies have investigated microbiome structure and rearrangement prior to PEx with inconsistent results[29,30,41,42]. Neither pathogen load nor other recurrent organisms were consistent predictors for imminent PEx across larger patient cohorts. Here, we stratified the microbiota by the identified PEx types and analyzed diversity and richness over time in trajectories with similar properties. Mixed effect models were implemented to test time dependencies of Shannon and Chao1 for the three PEx types and corrected for confounders (subject and PEx cycle) (Fig. 2B, C). All PEx types displayed significant diversity evolution across samples culminating in antibiotic treatment (LMM, $p_{PAT}$ = 1.26e − 3, $p_{AN1}$ = 4.46e − 2, $p_{AN2}$ = 3.67e − 2). The analog analysis for Chao1 identified PAT and AN1 to exhibit significant dependency with time (LMM, $p_{PAT}$ = 4.33e − 5, $p_{AN1}$ = 2.5e − 2).

Interestingly, richness and diversity decreased towards treatment for the pathogen-dominated communities (PAT) and increased for anaerobic PEx types (AN1 and AN2). Furthermore, it is important to note that the time dependency of richness and diversity was consistently small and ranged between $\eta^2$ = {0.02, 0.06} for all PEx types. In short, we revealed that diversity evolves oppositely, as the microbial communities approached PEx treatment. Of note, despite the modest effect size, these results have the potential to explain the inconclusive reports of previous studies that were conducted without consideration of PEx regimes[7].

**Species turnover displays antagonistic patterns in pathogen or anaerobe communities**

Evidence suggests that changes in airway microbial community structures may precede the onset of clinical symptoms of PEx by days or even weeks[15,41,43]. To determine the most likely time interval for such changes, samples were systematically grouped by collection time (days before the initiation of antibiotic treatment for PEx) and tested for significant differences in Shannon diversity and Chao1 richness (Fig. S5). A split into 1–23 and 24–60 days before PEx treatment showed statistically significant relative changes in all three PEx types, in accordance with the previous result indicating diversity and time dependency.

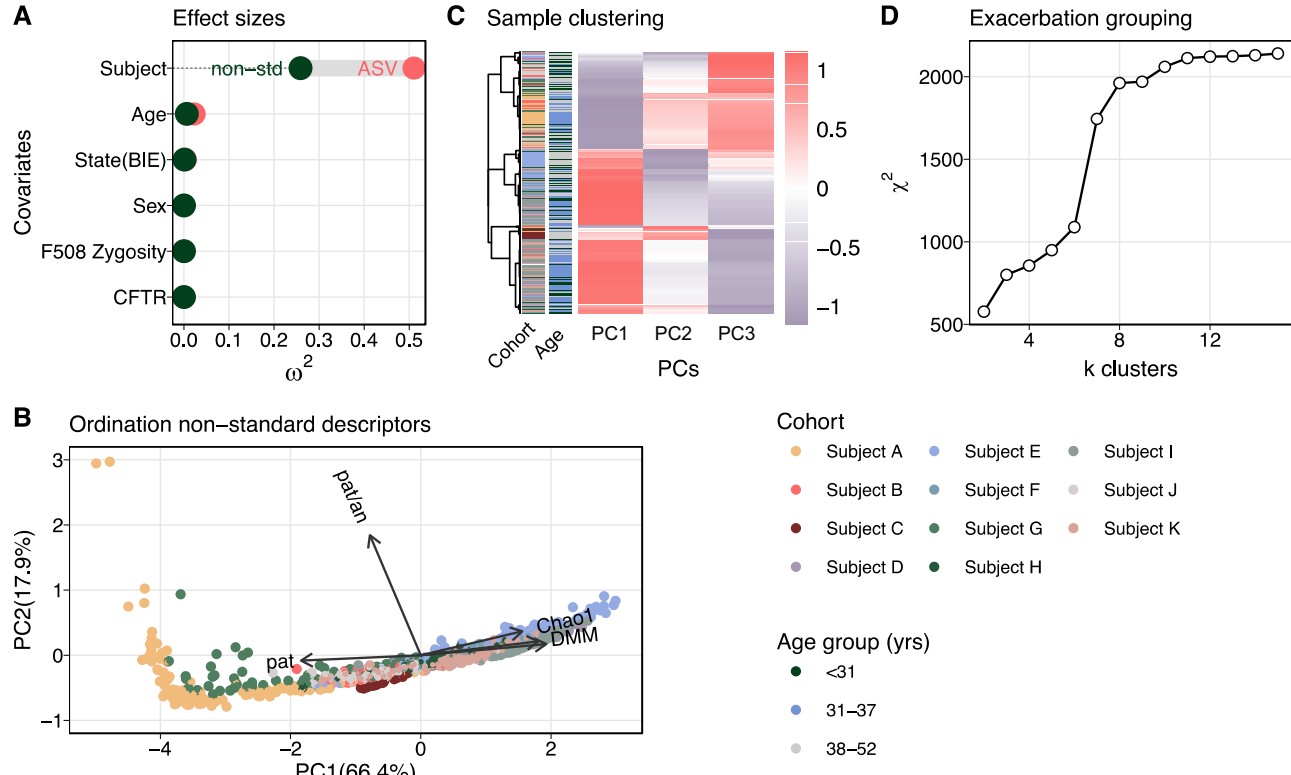

**Fig. 1 | Stratification of PEx types. A** Covariate bias explaining variance of microbiome data ($n = 880$ samples). Two PERMANOVA models contrasted the covariate effect sizes for ASV count data and non-standard sample descriptors. Partial values served as estimators of effect sizes $\omega^2$. Subject covariates included subject, age group (<31, 31–37, 38–52 years), clinical state (baseline, exacerbation), sex (female, male), F508del CFTR mutation zygosity (homozygous, heterozygous, n.a.), CFTR mutation (F508del$^{+/+}$; 3 groups F508del$^{-/+}$ and one other). **B** Principal component analysis using non-standard sample descriptors (explained variance = 84.3 %, two principle components (PCs) are shown). Model variables included Shannon diversity (Shannon), Chao1 richness (Chao1), relative abundance of the dominant pathogen

(pat), ratio of counts of CF pathogens and anaerobes (pat/an), and sample classification by Dirichlet multinomial mixture model (DMM) ($n = 789$ samples). Sample coloring by subject according to legend. **C** Sample-wise, hierarchical k-mer clustering and distance tree of ordinated data (PC1-3 indicate principle components). Subject ID and age group, as well as Pearson correlation coefficients of the samples are depicted for additional information. Color code of cohort and age group according to legend. **D** Identification of optimal k-mer number. The dependency between information gain and increasing cluster number $k$ is shown. First slope saturation served as a cutoff for the minimal number of clusters. Source data for Fig. 1 are provided in the Source Data file.

Community turnover $T$ describes the rate of species compositional change over time as defined by Ontiveros and colleagues[44]. We employed Aitchison distance to quantify community dissimilarity over time and assessed turnover as the slope of a fitted, linear model. We analyzed turnover $T$ during onset of (1–23 days prior to treatment) and prior to (24–60 days) PEx by evaluating Aitchison distance between any two samples collected in an interval of one to 20 days in a subject-wise manner (Fig. 2D). Overall, dissimilarity was smaller in the pathogen-dominated PEx type PAT (ANCOVA, p<0.001) and turnover $T$ was reduced during the 23 days compared to 24−60 days prior to PEx treatment (LMM, $T_{<24} = 0.17$; $T_{24−60} = 0.26$; p<0.001 for both tests). Interestingly, the anaerobic PEx types AN1 and AN2 again exhibited antagonistic patterns, with increased species turnover shortly before PEx treatment (ANCOVA, p<0.001 for both tests). Together, the previous results suggested two PEx regimes (PAT vs AN) with antagonistic temporal behavior.

## Characteristic community reorganizations stratify pulmonary dysbiosis types

The detailed organization of interactions and dependencies throughout an ecological community predefines its emergent, dynamical capabilities[45]. In particular, resilience to perturbations such as antimicrobial treatment, community robustness, and the stabilizing effect of keystone organisms were previously attributed to properties of dependency networks[18,19,22]. Therefore, it is not only important to identify the most relevant CF pathogen in the airway microbiome, but

to understand how the background community organization impacts the focal driver organism, modulates its virulence, and contributes to stability.

To study community organization, we inferred co-occurrence networks from sample subsets of individual PEx time series. In detail, for every network, 20 consecutively collected samples were used for robust inference[18] and a sliding window was employed to work across the individual PEx time series (with a step size of one sample). This approach yielded 589 co-occurrence networks, where topology changes between successive networks were caused by the substitution of a single sample. The resulting graphs were subsequently analyzed by PEx type ($n_{PAT} = 222$, $n_{AN1} = 192$, $n_{AN2} = 175$; detailed description in Table S2).

We studied the topology of the largest network components, defined as the ensemble of nodes belonging to the biggest connected subgraph of the network and, therefore, expected to be the most impactful for microbiome dynamics[46]. For PEx type PAT, a reduced number of organisms and associations were observed in the largest component, as well as increased betweenness centrality (Wilcoxon, p<0.001 for each pairwise test; Fig. 3A−C) in contrast to PEx types AN1 and AN2. Graph betweenness centrality measures the extent of centralized organization reinforcing effective communication patterns[47].

In analogy to interaction networks, we furthermore examined network hierarchy of the microbial co-occurrence networks across the PEx types. In the seminal work of Barabasi and Oltvai on biological interactions networks, a "quantifiable signature of network hierarchy"

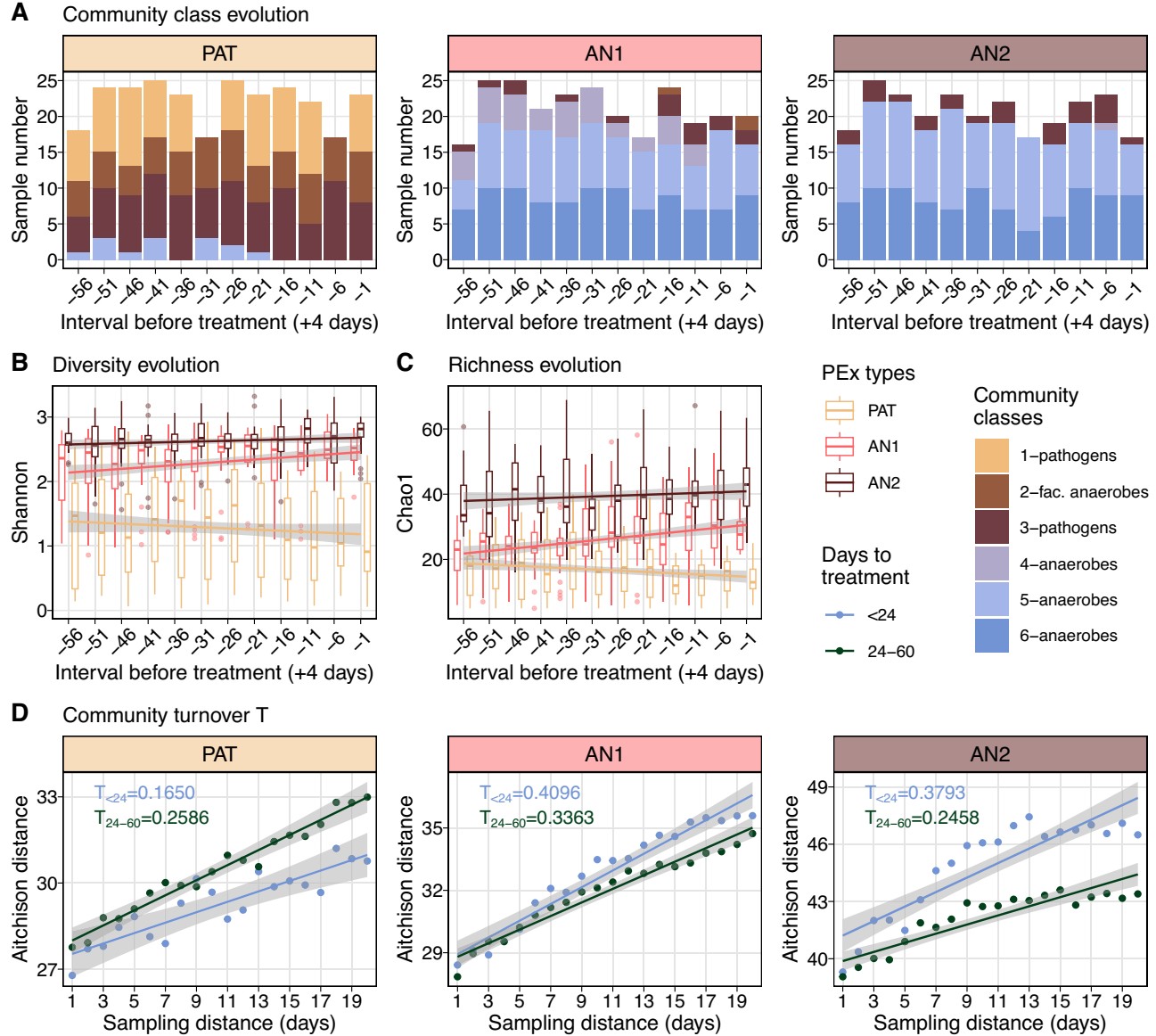

**Fig. 2 | Temporal behavior of PEx types. A** Time evolution of representative community compositions in three PEx types (PAT, AN1, AN2). Sample compositions were encoded by ASV groups and classified by a Dirichlet multinomial mixture (DMM) model into six community classes. Three classes were dominated by strictly anaerobic taxa, two by classical CF pathogens and one by facultative anaerobes (color coding according to legend). Community classes were independent of time towards start of PEx treatment but differed between PEx types ($n = 789$ samples). **B** Time evolution of microbiome diversity towards start of PEx treatment ($n = 789$ samples). Microbiomes in all PEx types showed significant time dependency (LMM, $p_{PAT} = 1.26e−3$, $p_{AN1} = 4.6e−2$, $p_{AN2} = 3.6e−2$) with Shannon diversity decreasing in pathogen-dominated and increasing in anaerobe-dominated PEx types towards treatment. **C** Time evolution of community richness towards PEx treatment ($n = 789$ samples). Chao1 richness changed significantly with time in pathogen-dominated and in one anaerobic PEx type (LMM, $p_{PAT} = 4.19e−5$, $p_{AN1} = 2.45e−2$). Boxes in

**B**, **C** depict the interquartile range; the horizontal line represents the median value. Whiskers extend to the minimum and maximum values, excluding outliers, defined as 1.5 times the interquartile range. **D** Community turnover $T$ relative to the previous 20 days for three PEx types. Samples were grouped according to their relative collection term prior to PEx treatment and mean values are plotted (in blue, mean of samples collected <24 days before treatment; in green, 24–60 days). Aitchison distance from focal sample to $n$ previously collected samples (sampling distance) was plotted, the slope of a linear fit was used to quantify $T$. Linear fit was represented with 95% confidence intervals of estimates (gray). $T$ was significantly reduced in pathogen-dominated microbiomes and significantly increased in anaerobe-dominated microbiomes in a 23-day-interval before PEX antibiotic treatment (one-sided ANCOVA for every PEX type, not shown in figure, $p_{PAT} = 2.07e − 8$; $p_{AN1} = 9.75e − 5$; $p_{AN2} <2e − 16$). Source data for Fig. 2 are provided in the Source Data file.

was defined as "the dependency of the clustering coefficient CC on the degree $k$ of a node, which follows $CC(k) \sim k^{-1}$"[48]. As a result, highly connected hub nodes should display low clustering, if located on top of the network hierarchy. To test this in co-occurrence networks, degree distributions $P(k)$ together with clustering distributions $CC(k)$ were inferred and compared by the slope of a power law fit across all co-occurrence networks within the same PEx type (Fig. 3D, E). While the fit to degree distributions showed similar slopes (LM,

$\alpha_{PAT} = −0.71$, $\alpha_{AN1} = −0.6$, $\alpha_{AN2} = −0.69$, $R_{PAT} = −0.64$, $R_{AN1} = −0.67$, $R_{AN2} = −0.73$, p<0.001), the slope of the node clustering distribution differed significantly between PEx types (LM, $\alpha_{PAT} = −0.97$, $\alpha_{AN1} = −0.27$, $\alpha_{AN2} = −0.28$). The pathogen-driven PEx type PAT displayed the strongest descent of clustering with degree k, was indeed approximating -1, and hence indicated a clear microbial hierarchy. These compelling results supported the hypothesis of a pronounced, hierarchical dysbiosis type. Moreover, the

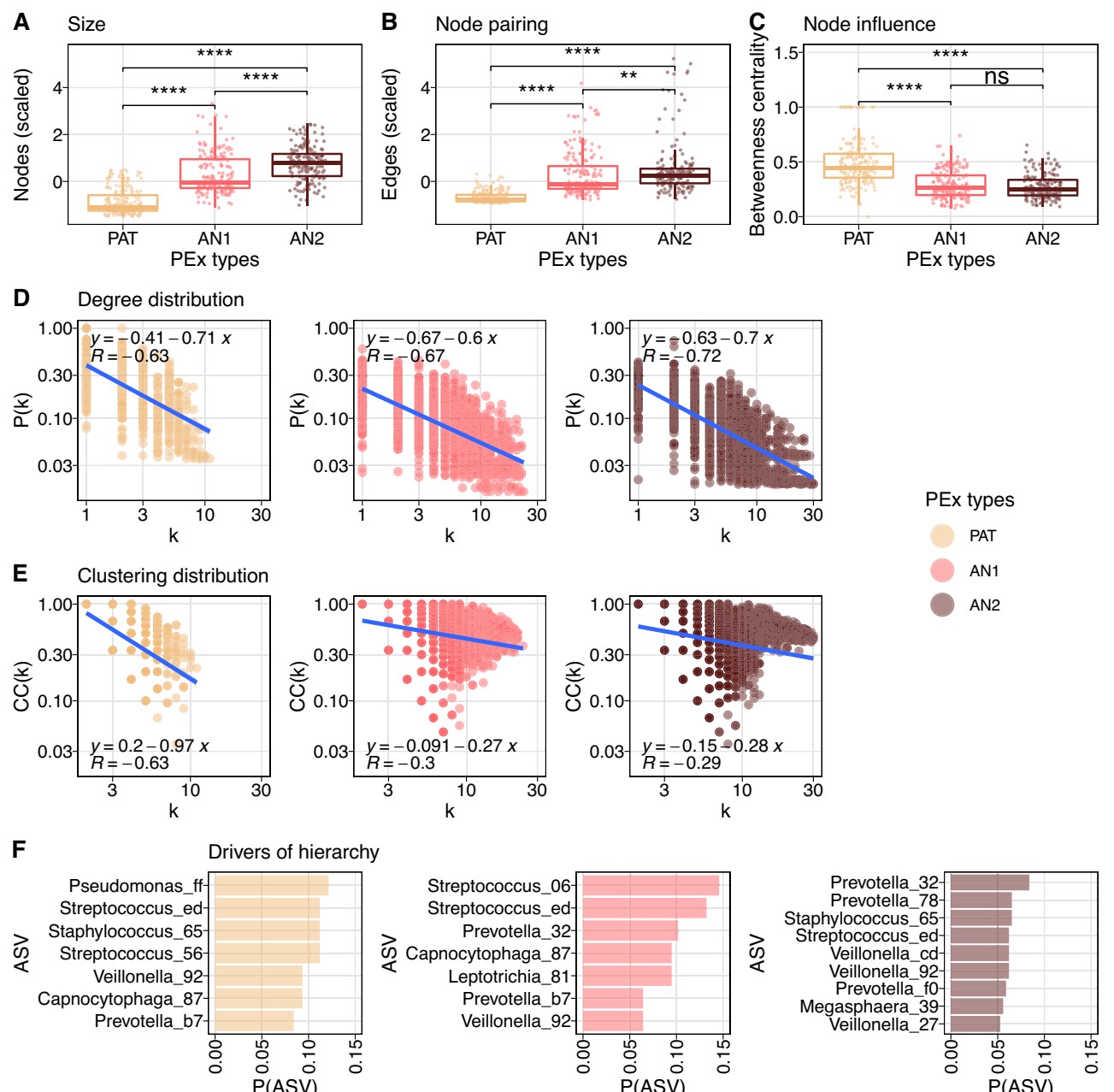

**Fig. 3 | Microbial community reorganization in pulmonary dysbiosis.** 589 co-occurrence networks were inferred by PEx time series, properties of the largest network component are depicted. **A** Boxplots of community sizes for three PEx types (PAT, AN1, AN2, $n = 589$). Pathogen-dominated communities were significantly smaller than anaerobic ones (two-sided Wilcoxon, $p_{PAT\ vs\ AN2}$, $< 2.2e - 16$, $p_{AN1\ vs\ AN2} < 1.7e - 9$, $p_{PAT\ vs\ AN1} < 2.2e - 16$). **B** Boxplots of edge numbers for three PEX types ($n = 589$). AN1 and AN2 displayed significantly more co-occurrences than PAT (two-sided Wilcoxon, $p_{PAT\ vs\ AN2} < 2.2e - 16$, $p_{AN1\ vs\ AN2} < 1.2e - 3$, $p_{PAT\ vs\ AN1} < 2.2e - 16$). **C** Information flow depicted as betweenness centrality for three PEx types ($n = 589$). Betweenness centrality in PAT microbiomes was significantly higher than in AN1 or AN2 communities (two-sided Wilcoxon, $p_{PAT\ vs\ AN2} < 2.2e - 16$, $p_{AN1\ vs\ AN2} < 0.52$, $p_{PAT\ vs\ AN1} < 2.2e - 16$). Boxes in **A**–**C** depict the interquartile range; the horizontal line represents the median value. Whiskers extend to the minimum and maximum values, excluding outliers, defined as 1.5 times the interquartile range. **D** Degree distribution over all co-occurrence networks

in respective PEx type. Probability $P$ of finding a network with degree $k$ in logarithmic scale; linear regression line depicted with model parameters; correlation of regression with data is reported. Power law in all three PEx types with comparable slopes $s = \{-0.71, -0.7, -0.6\}$. **E** Clustering distribution over all co-occurrence networks in respective PEx type (color code according to legend). The distribution of clustering coefficients $CC$ by network degree $k$ is plotted in logarithmic scale. Linear regression line is depicted with model parameters; correlation of regression with data is reported. Clustering for nodes with higher degree declined 3.5 times faster in pathogen-dominated organization than in anaerobe communities ($s = \{-0.97, -0.28, -0.27\}$, respectively). **F** Organisms residing in most hierarchical community positions. Top hierarchical positions were defined as nodes with highest 10% of all node degrees and lowest 10% of all clustering coefficients in the respective graph. $P$ indicates probability of ASV on this rank (minimal appearance cutoff $P = 0.05$) for three PEx types (color code according to legend). Source data for Fig. 3 are provided in the Source Data file.

anaerobic PEx types AN1 and AN2 also exhibited weaker correlation between fits and data (LM, $R_{PAT} = -0.61$, $R_{AN1} = -0.3$, $R_{AN2} = -0.29$, p<0.001) suggesting flat organization and unpronounced community structure.

To confirm that these results were driven by PEx types rather than sample diversity, we implemented independent linear mixed effect models for every graph readout, corrected for subject and calculated effect sizes of PEx types and covariates Shannon diversity and Chao1

richness. We found that betweenness centrality, clustering, number of vertices, and number of edges significantly depended on PEx types with effect sizes being 2.9 times, 5.5 times, 8.9 times and 3.4 larger than the most effective diversity measures, respectively (LMM, $p_{between} = 4.0e - 3$, $p_{cluster} = 4.8e - 2$, $p_{\#vertex} = 2.7e - 7$, $p_{\#edge} = 7.1e - 11$). Clustering and betweenness centrality were statistically independent of tested diversity measures, while both diversities influenced edge numbers and richness graph size to a minor extent (Fig. S6).

Next, we investigated which organisms preferentially occupied the most hierarchical positions in the communities and therefore likely controlled the overall microbiome dynamics. For each network, we identified the most hierarchical nodes, which were defined as nodes $i$ with a degree $k_i > 90\%$ and a clustering coefficient $CC_i < 10\%$ of all nodes in the graph. ASV frequencies were then assessed on these positions (Fig. 3F). In PAT communities, *Pseudomonas*, *Staphylococcus* and *Streptococcus* were not only masters in hierarchy, but also belonged to the most abundant ASV group in the samples. In AN2, a stronger variation of taxa was observed in the most hierarchical nodes. Of note, one third of the ranking was occupied by various ASVs belonging to the genus *Prevotella* in these communities. In Fig. S7, we visualized three representative PEx type communities with exemplary hierarchies using the Sugiyama algorithm for hierarchical graphs[49], which corroborated these findings. We concluded that in anaerobic communities, individual species were less relevant for overall microbiome dynamics, and microbiome organization was increasingly stochastic and less well picked up by co-occurrence analysis. On the contrary, in the pathogen-driven PEx type, the network hierarchies were conserved, occupied by a few key organisms and well supported by a simple, centralized community organization.

To contextualize these observations, we examined the identified configurations in contrast to microbiota in lung homeostasis. The definition of dysbiosis employed in this work defines a reorganization of the microbiota in the disease microenvironment. In healthy lungs, the pulmonary microbiome shows neutral community dynamics[50–52]. After Hubbell, diversity and abundance distributions of neutral communities can be explained by stochastic immigration and extinction events alone[53]. On the contrary, in chronic lung disease, microbial interactions, local replication, and environmental adaptation become key for diversity and community dynamics, and the impact of dispersal diminishes[13,50]. Consequently, if neutrality is a property of microbial eubiosis in the lung, then microbial interactions can be considered a hallmark of dysbiosis of the pulmonary microbiota. Together with metabolic adaptations, such interactions promote outgrowth of certain taxa to high relative abundances. Accordingly, we propose that both observed community states resemble different fundamental kinds of dysbiosis: the first, a structured, interacting community under the governance of an abundant, conventional CF pathogen, and the second, a globally successful functional guild that gains abundance by adapting to selective environmental pressures. Of note, similar community archetypes characterized by species-sorting or mass effects were described in metacommunity theory, a framework for ecological community assembly and dynamics[54]. The transition between the two metacommunity archetypes was explained by changes in dispersal due to altered spatial arrangements[38]. Here, we speculate that subject-specific mucus accumulation and decreasing oxygen availability in the lung microenvironment determine CF dysbiosis states in equivalent ways.

Importantly, both community states are robust maladaptations to the disease conditions of the lung, which raises the question whether negative loops exist in the system that enable their dynamical stability[13,55]. We hypothesize that in the first state, functional adaptations of the dominant pathogens together with antimicrobial defense against microbial competitors provide important negative feedback, whereas limitations of available niche space stabilize the second regime.

## CF pathogens drive PEx dynamics in hierarchical, but not in flat community organization

The virulence of pathogenic bacteria depends on microbial interactions and the biochemistry of the microenvironment among other factors. For example, *Pseudomonas aeruginosa* tightly regulates biofilm formation, as well as the production of siderophores and exotoxins based on iron availability and oxygen levels[56,57]. Moreover, the fermentation products 2,3-butanediol and lactic acid produced by anaerobic members of the CF microbiome were reported to trigger quorum-sensing and further virulence[58,59]. Conversely, synergistic interactions such as metabolic cross-feeding affect pathogen growth and lower the tolerance of *P. aeruginosa* to antimicrobial treatment independent of intrinsic antibiotic resistance profiles[60,61].

The insight that bacterial organization appeared markedly distinct in the identified PEx types raised the important question of whether microbiome organization could modulate pathogen importance or interfere with treatment outcomes in a foreseeable manner. As a first step, page rank was used as a statistical descriptor for network importance to compare the importance of conventional pathogens, strictly anaerobic, facultatively anaerobic, and strictly aerobic taxa in the community. We found that CF pathogens were differentially important for the CF community, displaying significantly higher page rank in hierarchical than in flat community organization (Fig. 4A). Next, pathogen dynamics in different community organizations were assessed using time series information. Previously, we demonstrated that stochastic ecological processes can be distinguished from interaction-driven processes by Fourier spectra inferred from the abundance changes of microbiota[62]. Here, the spectrum of every ASV per PEx time series was determined, and their noise color was inferred. White color indicated stochastic, whereas pink noise pointed to self-organized underlying processes. We observed that pathogens exhibited interaction-driven dynamics (pink noise) in steep hierarchies only (Fig. 4B). In flat anaerobe-dominated organization, pathogens instead showed stochastic behavior (white noise, AN2) or else anaerobic taxa dominated community dynamics (AN1). These results supported the hypothesis that CF pathogen activity depends on the community background and its positioning within (Fig. S7). Indeed, in graph theory it was demonstrated repeatedly that the topology of interaction networks was intimately linked with its dynamics[63–65]. Moreover, the particular configuration of microbial interactions has also been identified as a key element for robust composition forecasting[66]. In complex networks, interaction topology not only determined the importance of hubs for systems dynamics[20], but also controllability of dynamics supporting our conclusions[67].

To clarify the relevance of community organization for PEx treatment strategies, we asked whether community (network) organization could influence pathogen importance and modulate treatment outcomes. Recently, it was shown that dynamical fluctuation in response to external damage depended on the local network architecture of single-layered and multiplex networks[63–65]. In an analog, simplified approach, we modeled the response of our empirical co-occurrence networks to focal depletion of the dominant pathogen by antibiotic treatment. We hypothesized that community organization affected the degree of network disruption and, consequently, community dynamics and likely treatment outcomes. In the model, the most abundant pathogen was removed from the major component of 313 networks (NWs$_{PAT}$ = 152, NWs$_{AN1}$ = 53, NWs$_{AN2}$ = 108) and resulting network disruption was assessed by monitoring modularity change, breakup into subcomponents, and size reduction after single node removal (Fig. 4C–E). While total pathogen removal may not always be achieved in practice, microbial interactions were expected to abate together with strain abundance. Here we modeled the extreme case for the purpose of hypothesis testing. We found that pathogen elimination from steeper background hierarchies resulted in significantly stronger topology disruption supported by all three topological

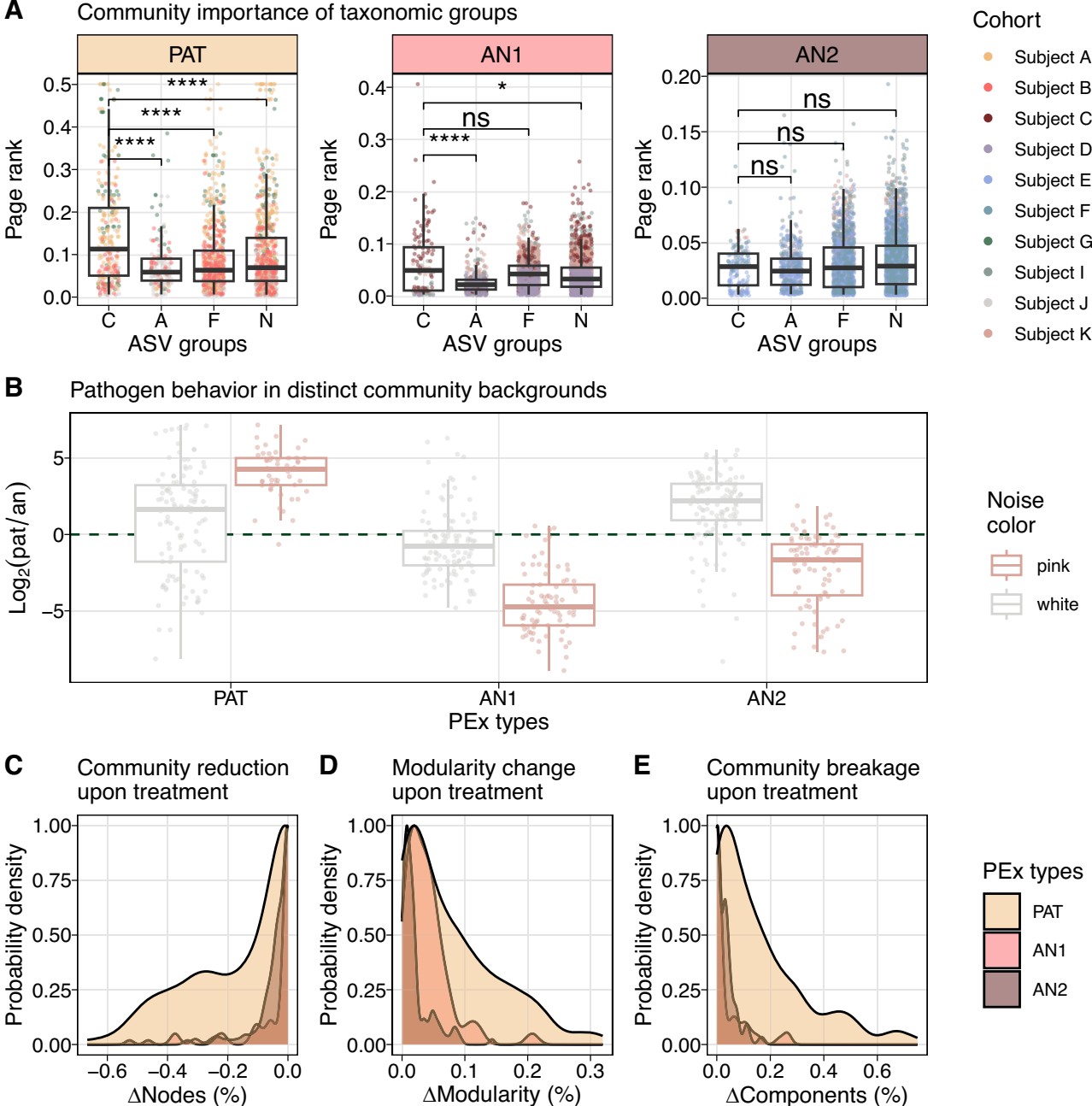

**Fig. 4 | Community organization modulates treatment success and defines the importance of CF pathogens. A** Community importance of organisms by ASV groups in different community backgrounds (C, CF pathogens; A, strict aerobes; F, facultative anaerobes; N, strict anaerobes). Page rank of nodes in co-occurrence networks for three PEx types (PAT, AN1, AN2) are presented ($n = 589$ networks). Pathogens are significantly more important than other ASV groups in pathogen-dominated microbiomes; aerobic community members are significantly less important in AN1 (two-sided Wilcoxon, PAT $p_{C \, vs \, N} < 7.4e - 8$, $p_{C \, vs \, F} < 6.1e - 11$, $p_{C \, vs \, A} < 2.8e - 9$; AN1 $p_{C \, vs \, N} < 3.0e - 2$, $p_{C \, vs \, F} < 3.1e - 1$, $p_{C \, vs \, A} < 1.8e - 6$). No significant differences were detected in AN2 (two-sided Wilcoxon, $p_{C \, vs \, N} < 8.4e - 2$, $p_{C \, vs \, F} < 3.4e - 1$, $p_{C \, vs \, A} < 2.2e - 1$). **B** Log2-change of noise colors generated by pathogenic and anaerobic organisms in three PEx types (PAT, AN1, AN2). Noise color was assessed by ASV, $n = 424$ were significant. The log2-ratio of pathogens to anaerobes (pat/an) associated with white or pink dynamics is plotted; green dashed line depicts shift of driver organisms (pathogens if $y > 0$; anaerobes if $y < 0$). White noise is the repercussion of stochastic time dynamics; pink noise the result of self-organized processes. CF pathogens displayed (self-)organized behavior in PAT only ($y_{pink} > 0$), and stochastic behavior ($y_{white} > 0$, AN2) or were irrelevant ($y_{white} < 0$, $y_{pink} < 0$, AN1) in anaerobic communities. Boxes in **A**, **B** depict the interquartile range; the horizontal line represents the median value. Whiskers extend to the minimum and maximum values, excluding outliers, defined as 1.5 times the interquartile range. **C–E** Simulations of targeted pathogen removal from data-derived co-occurrence networks. The most abundant pathogen in the sample was removed from the largest component and the resulting community perturbation was assessed ($n = 313$ networks). Probability densities of the effects are presented: **C** the relative size reduction (number of nodes) of the largest component; **D** the relative increase of modularity; and **E** the breaking of the largest component into smaller, unconnected components (relative difference of components). Hierarchical community organization in PAT is more strongly disrupted by focal treatment than is flat organization in AN1 and AN2. Source data for Fig. 4 are provided in the Source Data file.

parameters (Kolmogorov-Smirnov statistic in Table S3). We observed stronger change in modularity, increased disruption into unconnected subcomponents, as well as more pronounced size loss of the biggest connected component. Indeed, the depletion of the same organism resulted in divergent effects for the overall community architectures. The markedly different outcomes might serve as indicators for the degree of niche rearrangement after antibiotic treatment. We hypothesized that maintained niche accessibility should benefit repopulation after depletion, while niche reorganization could instigate the establishment of different community configurations. Although these data-derived hypotheses call for rigorous experimental testing, they are in line with previous clinical and experimental observations reporting altered responses of focal organisms to antibiotic treatment in different background communities[68,69]. In fact, response to antimicrobials may be recognized as an emergent property of the entire microbiome[70].

We concluded that the relevance of CF pathogens for microbial community dynamics and, by extension, likely also clinical course, was crucially shaped by community organization of the CF microbiome. Moreover, targeted treatment of pathogens resulted in distinct responses as a function of microbiome hierarchy, i.e., steep or flat community background.

### The importance of airway microbial community dynamics in CF

Our study on the human lung microbiome showcases the importance of community organization for understanding microbiome dynamics in homeostasis and dysbiosis. Using CF as a model disease, we employed functional/ecological coarse-graining to analyze both temporal and organizational aspects of pulmonary infection and its relevance for therapy outcomes.

We identified two archetypes of dysbiosis in the CF lung (driven by pathogens or anaerobes), characterized their community structures, and discussed stabilizing factors in the context of current graph and ecological theory. It is important to realize that the identified ecological features can cancel out if analyzed cross-sectionally due to their antagonistic nature. This might explain the difficulties with establishing robust predictors for PEx thus far[7].

We modeled the focal depletion of the most abundant pathogen in empirical co-occurrence networks and recognized that distinct background communities shaped the outcome of this treatment simulation. We concluded that the relevance of pathogenic taxa for microbiome dynamics, disease progression, and treatment effect is systematically linked to the organization of the background microbiome.

Our insights are limited to the ecological dynamics in airway microbiota observed in 11 adult subjects. Despite the large dataset investigated (880 samples, which comprise a tidy and comparable subset of a collection incorporating >21 patient years of near-daily sampling), this study cannot possibly reveal the full complexity of airway microbial ecology in CF. For example, more research is required to determine applicability to children with CF. Furthermore, none of the subjects in our study were receiving CFTR modulator therapy at the time of sample collection. How this therapy will impact CF airway microbiology is the subject of on-going studies. Nevertheless, we believe our model and the observations made in this study contribute to generating novel hypotheses regarding CF lung pathology, thereby building theory for targeted dysbiosis management, enhancing antibiotic stewardship, and advancing personalized medicine.

## Methods

### Study cohort, sample collection, and sample inclusion criteria

This observational prospective cohort study complied with all ethical regulations and was approved by the Institutional Review Board of the University of Michigan Medical School (HUM00037056) on March 17, 2011 (and renewed annually).

To obtain informed consent from participants, individuals who had previously expressed interest in participating in the study met face-to-face with a member of the study team before or after a regularly scheduled clinic appointment. The study team member reviewed the consent form with the individual in its entirety and answered any questions. Signatures of the individual and study team member were obtained on two copies of the consent form, one to go home with the individual and the other for study records. Five female and six male adult persons with CF were included in a balanced study design (sex determined by self-report) in this research. The study cohort is further characterized in Table 1. Expectorated sputum was collected as part of a long-term study of CF airway microbiota between 2011 and 2020. Subjects collected daily sputum samples at home, which were stored at either 4 °C for up to 35 days (mean 13 days), or −20 °C for up to 56 days (mean 25 days), before shipment to the University of Michigan on ice packs or dry ice for immediate storage at −80 °C. Electronic medical records were reviewed for subject demographic and clinical data. Among the 880 sputum samples and DNA sequences included in this study, 283 were reported previously (NCBI BioProjects PRJNA520924 and PRJNA611611)[24,25]. Sample inclusion criteria applied for downstream analyses are detailed in Tables S1 and S2.

### DNA extraction

Sputum samples were thawed on ice and homogenized with 10% Sputolysin (MilliporeSigma, Burlington, MA, USA). Samples were treated with bacterial lysis buffer (Roche Diagnostics Corp., Indianapolis, IN, USA), lysozyme (MilliporeSigma), and lysostaphin (MilliporeSigma) as previously described[31], followed by mechanical disruption by glass bead beating and digestion in proteinase K (Qiagen Sciences, Germantown, MD, USA). DNA was extracted and purified using the MagNA Pure nucleic acid purification platform (Roche Diagnostics Corp., Indianapolis, IN, USA) according to the manufacturer's protocol.

### Sequencing controls, protocol and taxonomic annotation

DNA libraries were prepared by the University of Michigan Microbial Systems Molecular Biology Laboratory. Human Microbiome Project (HMP) or Zymo (Zymo Research, Irvine, CA, USA) mock community standards were included on each sequencing plate. Negative water controls were included on each sequencing run, and reagent controls were prepared and sequenced with each new lot of reagents used in DNA extraction. In brief, the V4 region of the bacterial 16 S rRNA gene was amplified using touchdown PCR with barcoded dual-index primers (forward primer GTGCCAGCMGCCGCGGTAA, reverse primer TAATCTWTGGGVHCATCAGG[71]). Each PCR reaction was comprised of 1 μl of DNA template plus 4 μM equimolar primer set (5 μl), Accuprime High-Fidelity Taq (0.15 μl), 10× Accuprime PCR II buffer (2 μl), sterile PCR-grade water (11.85 μl). Touchdown PCR was performed consisting of 2 min at 95 °C, followed by 20 cycles of 95 °C for 20 s, 60 °C (starting from 60 °C, the annealing temperature decreased 0.3 °C each cycle) for 15 s, and 72 °C for 5 min, followed by 20 cycles of 95 °C for 20 s, 55 °C for 15 s, and 72 °C for 5 min and a final 72 °C for 10 min. The resulting amplicon libraries were normalized using a SequelPrep normalization plate kit (Life Technologies, catalog no. A10510-01), and concentrations measured using a Kapa Biosystems Library Quantification Kit (Kapa Biosystems, Wilmington MA) prior to being sequenced on an Illumina sequencing platform using a MiSeq Reagent Kit V2 (Illumina Inc., San Diego, CA, USA). The final load concentration was 4.0–5.5 pM with a 15% PhiX spike to add diversity, resulting in approximately 2x250bp reads (minimum depth 1031, average depth 15408.76 reads).

Annotation was performed using the dada2 pipeline in R according to the "Atacama soil microbiome" tutorial (https://docs.qiime2.org/2021.11/tutorials) using SILVA v138 for taxonomic assignments[72]. Samples were processed separately by subject through

sample inference to reduce batch effects introduced across multiple sequencing runs, then merged for the remaining processing steps. The dataset was denoised removing all ASVs with <0.0075% average abundance across all samples as previously described[73] and subsequently rarified using R package vegan[74]. Sequencing data, taxonomic information and clinical metadata were organized in phyloseq objects for further analysis[75]. Sequencing error rates were determined by comparing 43 mock community profiles to reference sequences in mothur (v1.48)[71] using the seq.error command, which measures error as the sum of mismatches to the reference divided by the sum of bases in the query. In 24 sequencing runs, the median mock community error rate was 0.037% (range 0.012−0.690%).

## Community typing with Dirichlet multinomial mixture models
To stratify representative community classes across subjects, Dirchlet multinomial mixtures (DMM) were inferred employing ASV groups as the taxonomic level[39]. Counts from ASVs were summarized into five groups: CF pathogens, strict anaerobes, facultative anaerobes, strict aerobes and unknown according to the oxygen requirements of the respective taxon[37]. Next, the total dataset was subject-stratified to avoid bias. Thirty-six random data subsets with 650 samples each were generated by sampling with replacement from the total data collection. Subsequently, models with 1-25 DMMs were inferred stepwise for each subset using the R package Dirichlet Multinomial[76]. Laplace approximation, BIC, and AIC were queried independently to identify the optimal number of DMMs.

## Variance testing and ordination
To quantify the impact of covariates on the lung microbiome, PERMANOVA was performed on rarified ASV data and the identical data were remodeled by non-standard sample descriptors[74]. Bray-Curtis distance was employed for ASVs, and Euclidian distance for scaled and centered non-standard descriptors. The model was designed to test the marginal effects of the individual covariates (function adonis2(), parameter setting by = margin). For comparison, effect sizes ($\omega^2$) of the covariates were calculated using the adonis_omegaSq() function from the MicEco package[77]. Principal component analysis was performed on scaled and centralized sample descriptors in R. Sample descriptors included Shannon diversity, Chao1 richness, relative abundance of the most abundant CF pathogen, the ratio of CF pathogen counts to counts from anaerobic taxa and the classification to a particular DMM community class.

## PEx type clustering and sample classification
Hierarchical k-mer clustering of samples was conducted on the first three principal components of the ordinated sample data using the package pheatmap[78]. Pearson correlation was employed as a similarity measure. Next, an $\chi^2$ contingency test was used to identify the best k for the classified sputum samples. Subsequently, entire PEx time series and networks were assigned to a single k-mer cluster by majority vote of the included samples. Two PEx time series were excluded from further analysis, because frequent type transitions prevented a conclusive association to a single PEx type. Detailed inclusion criteria are explained in Tables S1 and S2.

## Statistical modeling of time behavior
Linear mixed effect models to determine the time dependency of Shannon diversity and Chao1 richness were built using time groups (<24 days and 24−60 days before PEx antibiotic treatment; PEx types PAT and AN1) or 5-day intervals (PEx type AN2) as fixed effects and subject, age group, as well as PEx cycle as random effects (package lmerTest)[79]. Standardized effect sizes ($\omega^2$) of predictors were calculated using the package effect size[80].

ANCOVA models were implemented using time groups (<24 days and 24−60 days before PEx treatment) as categorical predictors,

sampling distance as numerical covariate, and Aitchison distance as dependent variable. Aitchison distances were calculated using R package robCompositions[81].

Graphical representations of boxplots and regression models were generated using ggplot2[82] and ggpubr[83].

## Co-occurrence network inference and network statistics
Co-occurrence networks were inferred from 20 samples collected on consecutive days with SparCC in a sliding window along each PEx time series[84]. Missing samples were imputed using R package seqtime[62] and jump size for the sample window was set to 1.

For downstream analysis on the largest network components, only strong ($\rho > |0.2|$) and statistically significant (p < 0.01 after FDR correction) co-occurrence edges were included, as well as networks with < 10 imputed samples. Topological properties were assessed using R package igraph[49].

Node degree and node clustering distributions were calculated across all networks classified in the same PEx type. To identify power laws and their respective slopes, linear regressions were performed on log-transformed data using R package ggplot2[82].

For comparing the effect size of PEx clusters with covariates Shannon and Chao1 on graph topology, we first calculated clustering coefficient, betweenness centrality, node counts, and edge counts for each largest component and averaged Shannon and Chao1 of all samples included for inference of the respective network. Next, we implemented independent LMMs, corrected for subject and assessed the effects sizes (partial $\eta^2$) as described previously.

To quantify the presence of certain ASVs in top hierarchical network positions, we first selected positions with the relative highest degree (>90% of all degree values) and relative lowest clustering (<10% of all clustering coefficients). Subsequently, the frequency of ASVs on these positions was counted, normalized, and ranked. All calculations were performed in R.

## Noise analysis of ASV time behavior
For each PEx time series, noise colors of participating ASVs were inferred using seqtime as previously described[62]. In short, missing samples were interpolated, and rounded to counts, negative interpolation values were set to 0 counts. Subsequently, the wrapper function identifyNoisetypes() performed a spectral density estimate and calculated a linear fit to the resulting periodogram (log frequency vs log spectral density) of the ASV time series. According to the slope of the fit, ASV time series were classified into categorical noise color groups. Noise colors were plotted as ratios of pathogens vs anaerobic ASVs with similar color (relative abundances) in the same sample.

## Pathogen removal
To identify the dominant pathogen by network, the subset of samples used to infer the individual co-occurrence network was queried and the pathogen with highest cumulative abundance was selected. Only networks with the dominant pathogen locating to the largest component were used for further analysis. We calculated modularity, the number of unconnected components and node size of the largest component independently for each co-occurrence network before and after pathogen removal. All parameters were normalized to the corresponding value before node removal. Density distributions of the normalized parameters were scaled and plotted with ggplot2 function geom_density()[82]. To test for difference of the cumulative parameter distributions, two-sided Kolmogorov−Smirnov tests were performed (ks.test(), stats package)[85].

## Statistics and reproducibility
The study data were collected in an exploratory, prospective cohort study design. No statistical method was used to predetermine sample size. Subject inclusion criteria are described in the main text; samples

included to individual analyses are detailed in tables S1, S2. The experiments were not randomized. The Investigators were not blinded to allocation during experiments and outcome assessment.

## Reporting summary

Further information on research design is available in the Nature Portfolio Reporting Summary linked to this article.

## Data availability

The generated sequencing data (FASTq files) are available as NCBI BioProjects under the accession numbers PRJNA987026, PRJNA520924, and PRJNA611611. Moreover, data are presented disaggregated by sex of the sample donor for every figure in the Source Data file. Furthermore, a list of ASVs as they appear in ASV groups and PEx types was added to Supplementary information/Source Data file. Source data are provided with this paper.

## Code availability

The R script collection and input data for this study are hosted at GitHub https://github.com/swidder/pex_types[86]. Large input data for code execution are hosted at Zenodo [https://zenodo.org/records/11109986].

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

## Acknowledgements

We gratefully acknowledge the individuals who provided the sputum samples used in this study. This work was supported by National Institutes of Health grants R01HL136647 and R56HL126756 and Cystic Fibrosis Foundation grants LIPUMA13IO and LIPUMA15PO (J.J.L.). SW was supported by the Austrian Science Fund (FWF) Elise Richter project V585-B31. We thank Ginestra Bianco for the critical discussion of the presented network models and Philipp Starkl for valuable feedback on figure design.

## Author contributions

S.W. designed the overall study and analyses. J.J.L. and S.W. engaged in the acquisition of the financial support for the study leading to this publication. S.W. and K.O. performed the data analyses. J.J.L., L.A.C., L.J.C., L.M.K. planned the daily sputum study, coordinated with participants, and collected samples, meta- and 16 S rRNA gene amplicon sequencing data. S.W., J.J.L., K.O., L.A.C., and L.J.C. participated in discussions related to this work. S.W. and J.J.L. wrote the manuscript. All authors read and approved the final manuscript.

## Competing interests

The authors declare no competing interests.
