## [Peer Review File · Nature Communications]

Microbial community organization designates distinct pulmonary exacerbation types and predicts treatment outcome in cystic fibrosisEditorial Note: This manuscript has been previously reviewed at another journal. This document only contains reviewer comments and rebuttal letters for versions considered at Nature Communications.

REVIEWERS' COMMENTS

Reviewer #1 (Remarks to the Author):

Having reviewed a previous version of this manuscript that was submitted to Nature Ecol Evol, I do appreciate the effort the authors have put into addressing my numerous concerns. The addition of key information previously omitted (e.g. including basic information such patient data, patients are adults, etc) and clarification of key concepts makes the manuscript more accessible to the reader than before.

I accept that the longitudinal data is hard won and valuable. However, I am of a strong opinion that it is the number of pulmonary exacerbations (PEX) and patients is the limiting factor here. Or rather 18 PEX from 11 adults with CF is not sufficient to support the main broad claims of this study, i.e. (1) that lung microbiota in PEX fall into one of two categories (CF pathogen led or anaerobe led), and (2) that this would predict treatment outcomes more broadly for people (children and adults) with CF. It is the case that data from more patients is required to support these interesting findings and to assert its generality within CF (of course to claim beyond CF would require sufficient equivalent sampling data from conditions such as COPD, severe asthma, non-CF bronchiectasis, etc).

Encouragingly, the authors state in their rebuttal that "The samples reported in this manuscript are part of a large multiyear project that has included sequencing of >7,000 CF sputum samples". I would politely suggest that revisiting those samples and data may help bolster the number of PEX and people with CF that could be included in the first instance. Although if it is to be claimed the overarching findings are applicable to anyone with CF (children and adults) then paediatric patient derived data would also be needed.

Reviewer #2 (Remarks to the Author):

The authors have made substantial changes to the manuscript that increase clarity and make it a very enjoyable read. No major comments remain; 2 small comments that the authors are open to address if they agree:

Fig S3A shows the data by subject yet the exclusion of data was done by timeseries; it would be more helpful to see the timeseries data in this Figure.

Line 363- I think this section would benefit from an additional sentence – similar to what is in the legend for Fig 4B – to explain white and pink noise.

Point-by-point response

Reviewer #1

Having reviewed a previous version of this manuscript that was submitted to Nature Ecol Evol, I do appreciate the effort the authors have put into addressing my numerous concerns. The addition of key information previously omitted (e.g. including basic information such patient data, patients are adults, etc) and clarification of key concepts makes the manuscript more accessible to the reader than before.

I accept that the longitudinal data is hard won and valuable. However, I am of a strong opinion that it is the number of pulmonary exacerbations (PEX) and patients is the limiting factor here. Or rather 18 PEX from 11 adults with CF is not sufficient to support the main broad claims of this study, i.e. (1) that lung microbiota in PEX fall into one of two categories (CF pathogen led or anaerobe led), and (2) that this would predict treatment outcomes more broadly for people (children and adults) with CF. It is the case that data from more patients is required to support these interesting findings and to assert its generality within CF (of course to claim beyond CF would require sufficient equivalent sampling data from conditions such as COPD, severe asthma, non-CF bronchiectasis, etc).

Encouragingly, the authors state in their rebuttal that “The samples reported in this manuscript are part of a large multiyear project that has included sequencing of >7,000 CF sputum samples”. I would politely suggest that revisiting those samples and data may help bolster the number of PEX and people with CF that could be included in the first instance. Although if it is to be claimed the overarching findings are applicable to anyone with CF (children and adults) then paediatric patient derived data would also be needed.

Indeed, a collection of >7000 sputum samples was the starting point of our analysis, and the subset of 880 samples included in this study represents the tradeoff between modeling feasibility and data availability. Although we sought to collect samples daily from a cohort of CF patients, collection patterns varied across and within subjects. The analyses we present required samples collected from each subject for a *minimum of 60 consecutive days without acute antibiotic treatment* and with relatively few missing samples. It was necessary to collect >7000 samples to obtain clean and comparable sets meeting these rather stringent criteria for longitudinal analysis. Of course, we would have welcomed including additional subjects, samples, and PEX, but no additional time series meeting these criteria were available. So unfortunately, we have no surplus data to include in or bolster our results. We agree completely that despite this enormous effort we are limited in how far we can generalize our findings. We have added text to highlight that our results are restricted to adults with CF; studies in larger numbers of pwCF, including children, are required to determine the applicability of these findings more broadly (L445-449).

Reviewer #2

The authors have made substantial changes to the manuscript that increase clarity and make it a very enjoyable read. No major comments remain; 2 small comments that the authors are open to address if they agree:

Fig S3A shows the data by subject yet the exclusion of data was done by timeseries; it would be more helpful to see the timeseries data in this Figure.

In figure S3A, we quantify and visualize the robustness of PEX type within individual subjects. In Figure S4B, the numbers of samples and their association with PEX clusters by time series are depicted. We feel that an additional figure like S3A for time series would therefore provide redundant information.

Line 363- I think this section would benefit from an additional sentence – similar to what is in the legend for Fig 4B – to explain white and pink noise.

We have added the following sentence for clarity (L374-5): “White color indicated stochastic, whereas pink noise pointed to self-organized underlying processes.”